# Justification of the Use of Composite Metal-Metal-Polymer Parts for Functional Structures

**DOI:** 10.3390/polym14020352

**Published:** 2022-01-17

**Authors:** Nickolay S. Lubimyi, Andrey A. Polshin, Michael D. Gerasimov, Alexander A. Tikhonov, Sergey I. Antsiferov, Boris S. Chetverikov, Vladislav G. Ryazantsev, Julia Brazhnik, İsmail Ridvanov

**Affiliations:** 1Department of Hoisting-and-Transport and Road Machines, Belgorod State Technological University Named after V.G. Shukhova, St. Kostyukov 46, 308012 Belgorod, Russia; info@polshin.ru (A.A.P.); mail_mihail@mail.ru (M.D.G.); cherep2240@rambler.ru (A.A.T.); await_rescue@mail.ru (B.S.C.); vladeslav390@gmail.com (V.G.R.); 2Department of Mechanical Equipment, Belgorod State Technological University Named after V.G. Shukhova, St. Kostyukov 46, 308012 Belgorod, Russia; anciferov.sergey@gmail.com (S.I.A.); rruzhaya@yandex.ru (J.B.); 3ERMAKSAN Makina San.ve TIC.A.S. Organize Sanayi Bölgesi, Lacivert Cad., Bursa 16065, Turkey; ismail.ridvan@ermaksan.com.tr

**Keywords:** metal polymers, composites, mechanical characteristics, generative design, additive manufacturing, topological optimization, material consumption, thermosetting plastic

## Abstract

The additive manufacturing of metal parts takes up an increasing number of areas of mechanical engineering, but it still remains too expensive for mass use. Based on the experience in the production of combined metal-metal-polymer forming parts of molds, a new method for the production of composite parts from a metal shell filled with metal-polymer is proposed. As a basis for the study, strength calculations are given by the finite element method for the details of the exoskeleton and a sample of simplified geometry. Comparison of the strength characteristics of parts made of various materials and their combinations showed high strength characteristics of a composite part made of a metal shell and a metal-polymer filler. The metal-metal polymer composite part is distinguished not only by its high strength but also by a significantly lower cost, due to the reduction in the volume of 3D printing with metal. The problems of obtaining composite structures are also discussed. The main problem is the development of a metal-polymer casting technology. The process of filling a thin-walled shell with a metal-polymer causes difficulty.

## 1. Introduction

The development of modern methods and technologies for the production of products is aimed at achieving the maximum manufacturability of the product. According to the definition of EASC (the Euro-Asian Council for Standardization, Metrology and Certification) 14.201-83, indicators of manufacturability of product design include such indicators as: labor intensity of product manufacturing, specific material consumption of a product, technological cost of a product, average operational labor intensity of maintenance and repair, average operational cost of maintenance and repair, average operational duration of maintenance and repair, specific labor intensity of product manufacturing, labor intensity of installation, coefficient of material applicability, coefficient of unification of structural elements, and collection coefficient. More generally speaking, manufacturability is a combination of product properties that ensure its lowest cost in design, manufacture, operation, and disposal. Within the framework of this study, such a parameter of manufacturability as the cost of production has been identified. The cost of manufacturing of a product depends, in turn, on the cost of designing a product, developing a technological process for its manufacture, the cost of equipment and tooling, a wage fund, etc., as a filler for a 3D printed casing.

Additive manufacturing is represented by various technologies, but for the production of loaded functional products, the most relevant technology for the manufacturing of metal products using the Selective Laser Sintering (SLS) method. The work [1] provides a compressed SWOT (Strengths, Weakness, Opportunities, Threats) analysis using metal 3D printing technology. It is necessary to highlight the main advantage of 3D printing with metals, this is a reduction in the weight of the product by optimizing the topology of the shape of the product, the absence of the need for technological support of production (equipment, tools). DMLS (Direct Metal Laser Sintering) and SLM (Selective Laser Melting) technologies proposed by German scientists also have their important drawbacks. In works [2,3,4], Craig Buchanan, Leroy Gardner, Yu-Lung Lo, Trong-Nhan Le, and others note that, in the process of 3D printing, porosity of the product and microcracks are formed when the powder is sintered with a laser, which reduces the density of the product and can lead to fatigue failure of the product. Although the report by Wohlers Associates [5] predicts an increase in additive manufacturing of $ 16–20 billion for all products and services, additive SLM technologies to produce loaded functional parts, even if the existing shortcomings are not taken into account, for many industries remain inaccessible due to the high cost of 3D printing, including the cost of metal powder, the cost of equipment, and the cost of post-processing. Table 1 shows the cost of printing 1 cm^3^ of stainless steel in various companies in November 2021.

Thus, the manufacturing of functional parts experiencing significant loads using the SLM technology is a rather expensive method, and it is used in limited areas of industry. Obviously, the price of SLS, SLM printing is a limiting factor for wider application of 3D metal printing in a wider range of industrial production. Therefore, the task of reducing the cost of obtaining functional parts, obtained using additive manufacturing, is an important and urgent task, the solution of which will contribute to a new round in the development of additive manufacturing.

At the same time, cheaper methods of additive manufacturing, such as FDM (Fused deposition modeling), are limited in application due to their low functional properties with respect to metals. In their work [11], Ksawery Szykiedans and Wojciech Credo investigate the mechanical properties of inexpensive 3D FDM and SLA (Stereolithography Apparatus) prints. As a result of the study [11], they conclude that the real properties of printed products differ from those indicated by the manufacturer of equipment and materials in a smaller direction, and they also depend on the nature of printing, filling, 3D printing modes. It was also noted that the parts obtained by FDM and SLA by 3D printing methods have anisotropy of properties, which is due to the layer-by-layer technology of material deposition. In [12], FDM additive manufacturing technologies are used to manufacture technological bushings and plugs, that is, for unloaded parts. The use of FDM technologies for the production described in [12,13] mass production is not advisable, so it is more rational to use the technology of molding thermoplastic polymers into metal molds.

In a study [14], Abid Haleem and Mohd Javaid provide an analysis on the application of additive manufacturing in various areas of manufacturing. The study shows that the most published mentions of additive manufacturing for functional products are in the medical sector and overtake the aerospace, automotive, architectural, research and reverse engineering industries.

One of the most frequently cited industries in medical additive manufacturing is exoskeleton manufacturing. For example, in [15], parts of a training exoskeleton are made using 3D printing for the recovery of people after a stroke. Such parts are not loaded and FDM 3D printing technology is quite suitable for this. In addition, in [16], a device for fixing the limbs when recovering from an injury is proposed; parts of this device also do not experience serious stress and can be made of plastic. A completely different situation develops when the design of an exoskeleton for medical or industrial purposes is subjected to significant loads from the moving loads, the mass of the exoskeleton itself, dynamic loads, and the weight of the operator or patient. The study [17] provides examples of 3D printing in the manufacture of lower limb exoskeletons. Kamila Batkuldinova, Anuar Abilgaziyev, Essam Shehab, and Md. Hazrat Ali note that the use of metal 3D printing is expensive, and printing from polymers is characterized by low strength and possibly more cumbersome since, to compensate for the low strength of polymer printing, it is necessary to increase the size of the material layer.

Nowadays, the market of industrial metal-polymer materials is widely represented by metal-polymer materials with close physical and mechanical properties to metal, possessing unique flow properties at low temperatures. These casting properties make it possible to use these materials in the development of new production technologies, partially or completely replacing metallic materials. For example, in studies [18,19,20], it is shown that the use of a metal polymer is optimally suited for the manufacture of shape-forming tooling for molding thermoplastics in small-scale production. The metal polymer is able to withstand the temperature and pressure of the plastic melt injected into the mold, providing the required shape and size of the product.

Thus, on the one hand, there is an increasing need for the additive manufacturing of highly loaded functional products, but this is hampered by the high cost of 3D printing with metal. On the other hand, there are metal-polymer materials with unique strength properties. By combining the best from additive manufacturing technology—simplicity of design, high production speed, freedom of form production—and from the properties of metal-polymers—strength and flow properties—it is possible to develop an effective and economical technology for the production of composite parts. When using 3D printing, a thin-walled shell can be produced using cheap methods of printing with thermoplastic polymers, and after filling the cavity in this form with a metal polymer, a functional product that can withstand high loads, comparable to metal parts, will be made.

The purpose of this study is to substantiate the effectiveness of using a composite part, consisting of a metal-polymer filler and a thin-walled shell, made by various additive methods. Substantiation of the effectiveness of the application is based primarily on a comparison of the strength characteristics of composite parts and parts completely manufactured by 3D printing methods with various materials. The economic efficiency of using a composite part, in comparison with an all-metal part, is substantiated by means of a calculated confirmation of the part’s strength. Confirmation of the strength of a composite structure, which contains a metal-polymer, will justify the possibility of replacing the internal metal filling of a part with a cheaper metal-polymer one. When comparing the strength characteristics of composite parts and parts printed with ABS (acrylonitrile butadiene styrene) plastic, superior strength characteristics of the composite part are expected over a part made entirely of ABS plastic.

Ultimately, the results of the study are aimed at confirming the relevance of the development of a technology for the manufacturing of composite parts, including the development of appropriate technological support, the selection of technological modes for casting metal-polymers, and substantiation of the modes of mechanical processing of composite parts.

## 2. Materials and Methods

The basis of the study, in describing the features of the technology for the manufacture of composite products, is the strength calculation by the finite element method of different variations of composite parts. Licensed software Solid Edge (221.00.04.003 × 64, Femap 2020.1.1 city Belgorod, Belgorod region, Russia) has been used to prepare part models and calculate them. The search and borrowing of models of parts for calculations has been carried out in the free access library GrabCAD community.

The physical properties of metal-polymer materials are analyzed on the basis of data from various world manufacturers. In addition, the study has taken into account the results of experimental studies obtained earlier in the development of technology for the manufacture of metal-metal-polymer molds [18,19,20].

Table 2 shows the properties of metal-polymer materials according to the manufacturers’ data; in the future, these characteristics will be used in the finite-element modeling of a composite product.

The model of the exoskeleton in Appendix A [25] lever presented in Figure 1 has been selected as the object of research.

It should be noted that this exoskeleton model was designed in the framework of NASA’s mission: a general containment and mobility system in conditions of multiple gravity. NASA challenged the GrabCAD community to develop a general containment and mobility system that works in four gravity environments. Among others, the exoskeleton model presented in Figure 1 has been chosen as the object of research, since the components of the presented model are best suited for considering the problem of their manufacturing using additive technologies.

Since the study involves a comparison of the strength characteristics of products made of metal, plastic, metal polymer and composite-metal-metal polymer parts, Table 3 shows the characteristics of the materials used in the calculations. Specifications are based on material manufacturer data and SolidEdge reference guides.

It can be difficult to study a composite part made of a thin-walled shell 2 mm thick and a filler in the form of a metal-polymer composition. A thin-walled shell of a part with a complex geometric shape can lead to inconsistency in the calculation results. For this reason, to study the composite specimen, a simpler geometric model of the part has been adopted, as shown in Figure 2. In Appendix A [26].

## 3. Results

The loading of the parts of the exoskeleton during calculations should be carried out by those loads that its structure can perceive in reality. The service purpose of the exoskeleton assumes that its parts are subject to total loads W associated with human weight HW, own weight OW, as well as loads from drives DL (drive loads). In general, the total loads W can be written by Equation (1):W = HW + OW + DL(1)

According to the mass-centering characteristics of the model, the average weight of a person, and the tested loads of the drives, the elements of the exoskeleton do not experience loads exceeding 2000 N or 200 kg. Based on this, the following models have been calculated and the results have been obtained.

### 3.1. Calculation of a Metal Lever

Figure 3 shows the calculation and diagrams of stresses, deformations and safety factors for a lever made of metal. The lever has been fixed along the lower hole, and 2000 N loads, directed to the upper lugs, were perpendicular to the axis of the lever, that is, the lever works for bending. Material characteristics have been specified in accordance with Table 3 “Stainless steel”.

The calculations presented in Figure 3 show that the lever made of metal withstands the specified operating conditions. The safety factor allows it to be loaded with loads 2.4 times higher than the existing ones. The maximum load value is 154 MPa, while the tensile strength of the metal lever is 610 MPa. The maximum deformations correspond to 0.555 mm. Perhaps this indicator may affect the performance of the structure, since elastic deflection occurs under the action of the created torque. However, in real conditions, the lever will not experience the full rated load, since it will be distributed throughout the entire structure of the assembly unit. Therefore, this value of deformation can be neglected.

### 3.2. Calculation of the Metal-Polymer Lever

Figure 4 shows the calculation and diagrams of stresses, deformations, and safety factors for a lever made of metal polymer. Lever fixation and load are similar to the calculation of a metal lever. The characteristics of the material were specified in accordance with Table 3 Metal polymer “Ferro-chromium”.

An analysis of Figure 4 shows that a metal-polymer lever experiences the same maximum load as a metal lever. The value of deformations due to elastic deflections can reach 19.8 mm, which seems to be quite significant. However, as indicated earlier, this value must be considered weighed, in the context of the actual operating conditions of the lever. The safety margin of a metal-polymer lever makes it possible to judge whether the lever can be loaded by 0.426 more than the current load and this will not lead to its destruction.

### 3.3. Calculation of the Lever Made of ABS Plastic

Figure 5 shows the calculation and diagrams of stresses, strains, and safety factors for a lever made of ABS plastic used in FDM 3D printing. Lever fixation and load are similar to the calculation of metal and metal-polymer levers. The material characteristics have been specified in accordance with Table 3 “ABS plastic”.

Analysis of Figure 5 shows that a part made of ABS plastic undergoes significant deformations up to 73 mm. In combination with the fact that a layer-by-layer grown part has a lot of defects in the form of non-adhesions, pores, and a difference in the printing layer [11], and in calculations, the model has a solid structure. It can be judged that this part will not be able to withstand the given loads. The safety factor, in this case, although it has a positive value of 0.267, is also true for a solid part made of ABS plastic—for example, if the part has been cast, and not printed on a 3D printer.

### 3.4. Calculation of the Strength of Samples Made of Metal, Plastic and Metal-Polymer

Further, the calculations of the sample part will be given, and its geometric characteristics are shown in Figure 2. All calculation parameters are saved for the model of the exoskeleton lever. Calculation on a simplified model of the sample has been introduced into the study in order to simplify the model, which leads to more accurate mesh generation and calculation. Calculations using the example of a sample are intended to show a visual comparison in the strength characteristics of various materials, which can be transposed into more complex parts, e.g., parts of an exoskeleton.

Figure 6 shows the calculated values for a metal sample.

Figure 7 shows the calculated values for a metal-polymer sample.

Figure 8 shows the calculated values for an ABS plastic sample.

The calculations of the exoskeleton lever model Figure 1b fully correlate with the calculations of the sample Figure 2, which allows us to judge the adequacy of further calculations of composite structures, using the example of the sample model Figure 2.

Further, the data on the calculation of composite samples of metal-metal-polymer and ABS plastic, namely metal-polymer, will be presented.

### 3.5. Calculation of Composite Parts

Figure 9 shows a longitudinal section of such a shell. Figure 10 shows the calculation and diagrams of stresses, strains, and safety factors of a composite sample made of a thin-walled metal shell 3 mm thick and filled with a metal-polymer (metal-polymer not shown).

Figure 11 shows the calculation and diagrams of stresses, deformations, and reserves of strength of a composite sample made of a thin-walled plastic ABS shell, 3 mm thick and filled with a metal-polymer.

## 4. Discussion

### 4.1. Analysis of the Calculated Data

For more convenience in discussing the obtained results, let us summarize the results of all calculations into a single diagram, presented in Figure 12.

The stress diagram, Figure 12, shows that the highest stresses are experienced by a composite sample of a metal shell and a metal-polymer, which is explained by the difference in the tensile strength of the two materials, a result of which stress concentrators appear, as shown in Figure 13. While designing composite parts and their preliminary calculation, the locations of possible stress concentrators should be taken into account. To reduce stresses in such places, it is necessary to increase them by increasing the layer of metal material.

The deformation diagram Figure 12 shows that the largest deformations are experienced by the ABS plastic sample and the ABS plastic, which is a metal-polymer composite sample. This is natural, since ABS plastic has the lowest Young’s modulus in relation to other materials. According to these calculations, from the point of view of the applicability of this material in functional structures, this is a critically negative factor, since a violation of the geometry of the product design will lead to the inoperability of the entire mechanism. In this case, a composite sample made of a metal shell and a metal-polymer filler is the least susceptible to deformation after an all-metal sample. In addition, the difference in deformations in these two cases is the smallest among others. This indicates the priority of the use of such a combination of materials in the design of various structures, since the deformations are still insignificant, and the economic effect of the use of a metal-polymer is very noticeable.

Analyzing the safety factor diagram, Figure 12, it becomes clear how many times the load on a part made of a particular material or their composition can be increased without destroying the part itself. The safety factor diagram, Figure 12, shows that the metal specimen has the greatest safety factor. The second, in terms of safety margin, is a composite sample of a metal shell and a metal-polymer filler, followed by a completely metal-polymer sample. The fourth, in terms of safety margin, is a composite sample made of a shell (ABS plastic) and a metal-polymer filler. The least durable of the tested samples is a sample made of pure ABS plastic.

In accordance with the purpose of the study, it is necessary to substantiate the calculation of the strength of a composite part consisting of a metal-polymer filler and a thin-walled shell made by various additive methods. The safety factor of a specimen made of a metal shell and a metal-polymer, when loaded with 2000 N, has a safety factor of 3.75, which means that it can withstand a load of up to 7500 N or about 750 kg. Therefore, the conclusion is that it is possible to use a part that is not completely printed with metal on a 3D printer, but a composite part consisting of a metal shell and a cheap metal-polymer filler, which will multiply the economic efficiency of the production of such parts.

### 4.2. Problems of the Implementation of the Technology for the Manufacture of Composite Parts

At first glance, the calculations performed show the obvious advantages of using a composite structure of parts made of a metal shell filled with a metal-polymer. Such parts have a high margin of safety, small deformations, and a relatively low cost, since the cost of 1 cm3 of metal-polymer [23] equals $ 0.01125, which is 1136 times cheaper than making 1 cm^3^ of a metal part printed on 3D printer. However, today, there are no technologies or technological support for the manufacture of such parts, since there are a number of restrictions:Density of distribution of metal-polymer in the form;Rheology of metal polymer in mold channels;Thermal destruction of the metal polymer during abrasive processing;Lack of technological support for pouring metal polymer into a mold.

#### 4.2.1. Density and Distribution of Metal Polymer

The calculation by the finite element method of the strength of the composite structure of a part during its design is carried out on the basis of the condition of complete filling of the form with a metal polymer. That is, it is possible to judge the strength characteristics of a composite part only under the condition that there are no underfills, air pockets and pores in the form, which can cause the formation of internal stress concentrators and a decrease in the strength of the parts. Early experiments in the production of metal-polymer forming parts of molds [27] showed that, when mixing the components of the metal-polymer, the viscous-flowing mass is saturated with gaseous inclusions. Figure 14 shows an enlarged cut of a metal-polymer mass cured at atmospheric pressure. The study of the microstructure of the metal-polymer sample has been carried out using a high-resolution scanning electron microscope TESCAN MIRA 3LMU, according to the method [28].

Figure 14 shows the presence of a mass of air bubbles, the diameter of which reaches 0.5 mm. That is why metal-polymer manufacturers do not recommend applying a metal-polymer layer when repairing parts over 10 mm. Therefore, when filling hollow shells with metal-polymer polymers, it is required to ensure a high density of the metal-polymer, which is an unsolved problem.

#### 4.2.2. Metal-Polymer Rheology in Mold Channels

Another limiting factor in the use of a metal-polymer in the proposed technology for manufacturing composite parts is the rheological properties of a metal-polymer. Since the viscosity of metal-polymers range from 20,000 mPa × s to 25,000 mPa × s, in certain cases, the size of the channel cross section through which the metal-polymer flows when filling the mold may be insufficient, which will also lead to underfilling and cavities in the mold when hardening.

Understanding the relationship between the pot life of a metal-polymer and its rheological properties when filling a mold with a variable cross-section of channels is an important aspect when designing and optimizing the shape of a product using generative design [29,30,31,32,33], which makes it possible to minimize the material consumption of a product and minimize the cost of manufacturing a product.

In studies devoted to the rheology of polymers [34,35,36,37,38], it is indicated that the vibration excitation application, in the process of casting polymers, significantly reduces the viscosity of the polymer composition, in some cases [36] up to 9.6 times. However, these studies relate to thermoplastic polymers, and additional studies are required for thermosets.

#### 4.2.3. Thermal Destruction of Metal-Polymer during Abrasive Processing

The use of a metal shell carries, in itself, the function of not only imparting rigidity to the shell, but also providing the required properties of the working surfaces. For example, bearings or pins are installed in the lugs of the exoskeleton lever Figure 1b, which ensure the rotation of the hinge; therefore, for such surfaces of parts it is necessary to perform additional machining. Most often, this is grinding.

Studies in the production of combined parts of molds [39,40] have shown that, during abrasive processing of two materials, metal and metal-polymer, a situation may arise when the metal-polymer may undergo temperature degradation. The fact is that, while processing a composite part consisting of a metal and a metal-polymer, it is the metal surface that is most often subjected to processing. The metal-polymer is usually located inside the structure. The temperature resistance of the metal-polymer reaches 280 °C. Grinding temperatures on metal surfaces can be up to 800 °C. Therefore, the internal metal-polymer can be heated by the external metal (Figure 15) reaching temperatures exceeding its temperature resistance. This can lead to thermal destruction of the metal-polymer part of the composite part. A composite part in which the inner part will be destroyed by temperature will be unusable [41,42,43,44].

#### 4.2.4. Lack of Technological Support for Pouring a Metal-Polymer into a Mold

The difficulties of casting a metal-polymer specified in Section 4.2.1 and Section 4.2.2 require the development of appropriate technological support to eliminate these factors in the production of composite metal-metal-polymer parts. To date, such a provision is not available on the market of technological equipment, although, in the first approximation, based on the existing production experience, a vibro-vacuum casting installation is presented. When the metal shell provides for the lower supply of the metal-polymer into it, the shell itself is placed in a vacuum chamber and there, it is fixed, while the vacuum chamber is located on the vibrating table. Further, through a flexible sealed pipeline, the liquid metal-polymer is supplied into the shell, while the vibration intensifies the flow of the metal-polymer, and due to the vacuum in the shell, the metal polymer is pulled into it, as it were. Thus, underfilling and pores in the metal-polymer part of the composite part should be eliminated.

## 5. Conclusions

The study consisted of comparing the safety factor of parts of the same shape, under the same load, but made of different materials, such as metal, metal-polymer, ABS plastic, as well as their compositions: metal-metal-polymer and ABS plastic–metal-polymer. In the first case, for clarity of practical application, a model of an exoskeleton lever was used as models of parts, and in the second case, a model of a sample, with dimensions close to the model of an exoskeleton lever was used. The sample model was used to improve the convergence of the calculation results, since it has simpler geometric parameters, which means that the accuracy of calculations when using a larger mesh of finite elements will be higher. This saved computing resources.

Calculated data showed that the design of a composite image, consisting of a thin-walled metal shell (made by the additive method) filled with a metal-polymer composition, under conditions of a given load equal to 2000 N, has a rather large margin of safety of 3.75 (Figure 12). This margin of safety makes it possible to judge that the proposed design of the composite part can be used under more loaded conditions while maintaining its performance.

It is necessary to say that, while using a composite structure consisting of a thin-walled metal shell filled with a metal-polymer composition, instead of a solid metal part, the volume of the metal volume of the part reduces. It is assumed that both the metal shell and the fully fabricated metal part are 3D printed with metal. As shown in Table 1, 3D metal printing is an expensive manufacturing method. Consequently, reducing the volume of 3D printing can significantly increase the economic efficiency of the manufacturing of functional parts, such as an exoskeleton lever.

At the same time, as shown by the strength calculations of the samples by the finite element method, the composite part has a sufficient margin of safety in order to withstand the loads perceived by it during operation.

The use of the composite structure ABS plastic–metal-polymer also has prospects of use, since 3D printing with plastic, for the manufacture of shells, is more accessible for production than 3D printing with metal, and the strength properties of the metal-polymer are higher than those of ABS plastic.

The scope of application of the proposed composite structure, consisting of a thin-walled shell filled with a metal-polymer, is limited only by the imagination of the design engineer.

On the basis of the available experience, the main problems of obtaining composite structures have been described. The main problem is the development of a metal-polymer casting technology. The complexity is caused by the process of filling a thin-walled shell with a metal-polymer-polymer, which can have a different cross-section in volume. The future research can help to form a whole complex of results that allow us to determine the criteria for the applicability of the technology, technological modes of casting, mechanical processing of a composite part, and to develop the appropriate technological support.

## Figures and Tables

**Figure 1 polymers-14-00352-f001:**
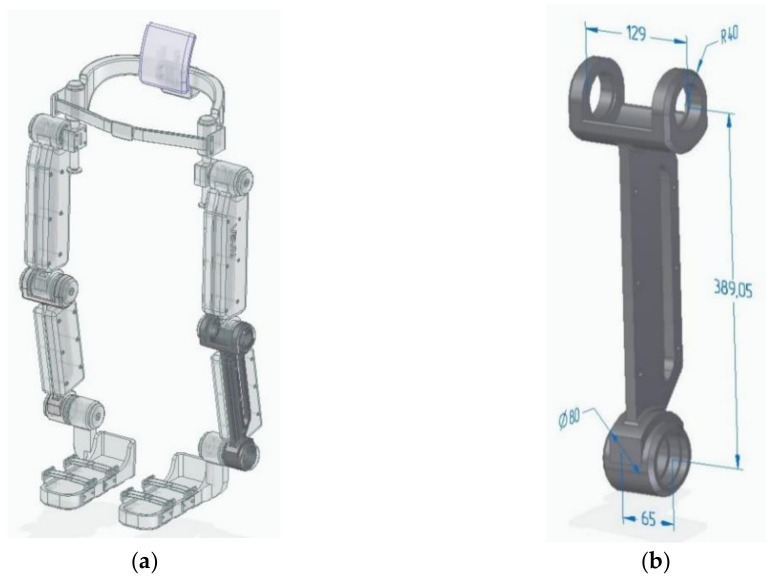
NASA exoskeleton model: (**a**) Lever highlighted in exoskeleton assembly; (**b**) Model of an exoskeleton lever.

**Figure 2 polymers-14-00352-f002:**
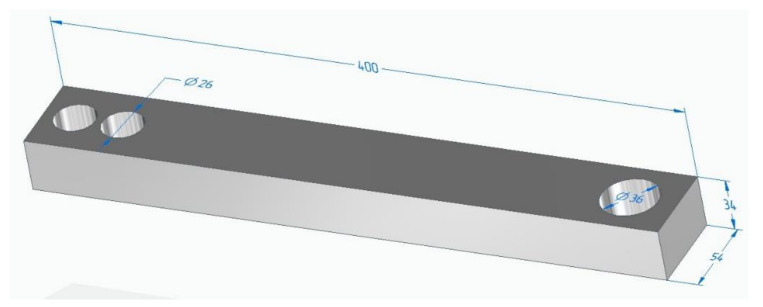
Simplified model of the sample for calculations.

**Figure 3 polymers-14-00352-f003:**
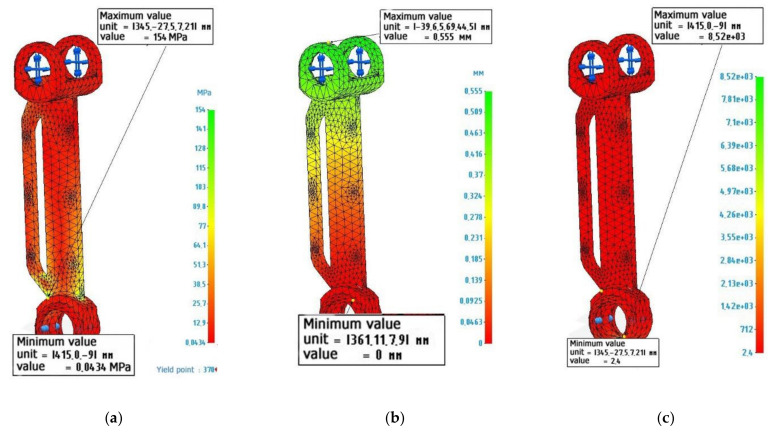
Finite element design of an exoskeleton lever made of stainless steel: (**a**) Von Mises stress diagram, maximum value 154 MPa; (**b**) Strain diagram, maximum value 0.555 mm: (**c**) Safety factor diagram, minimum value 2.4.

**Figure 4 polymers-14-00352-f004:**
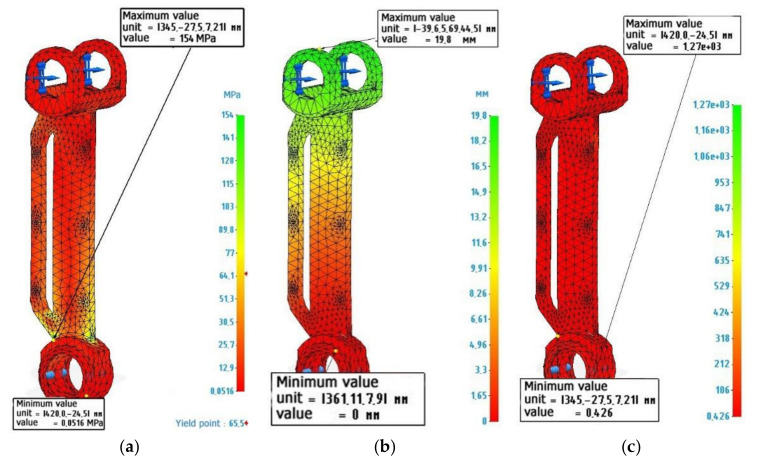
Finite element calculation of an exoskeleton lever made of Metal polymer “Ferro-chromium”: (**a**) Von Mises stress diagram, maximum value 154 MPa; (**b**) Strain diagram, maximum value 19.8 mm: (**c**) Safety factor diagram, minimum value 0.426.

**Figure 5 polymers-14-00352-f005:**
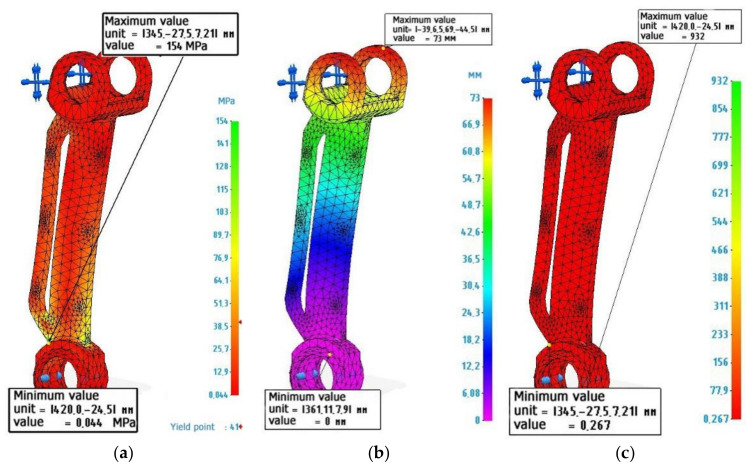
Finite element calculation of an exoskeleton lever made of ABS plastic: (**a**) Von Mises stress diagram, maximum value 154 MPa; (**b**) Strain diagram, maximum value 73 mm: (**c**) Safety factor diagram, minimum value 0.267.

**Figure 6 polymers-14-00352-f006:**
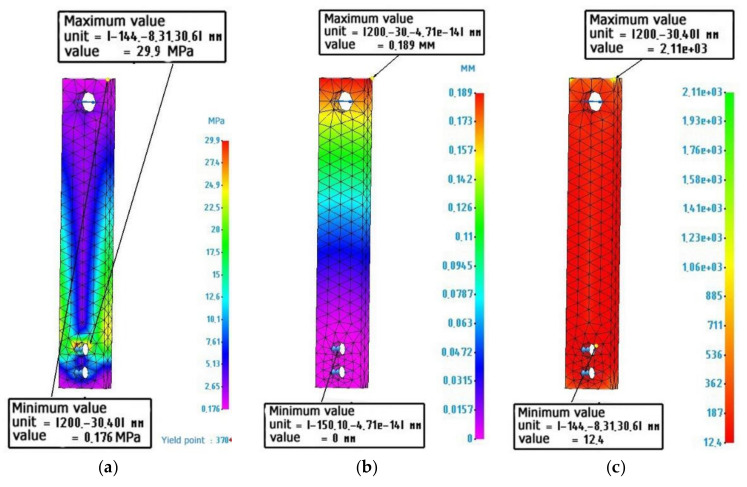
Finite element analysis of a metal sample: (**a**) Stress diagram according to von Mises, maximum value 29.9 MPa; (**b**) Strain diagram, maximum value 0.189 mm: (**c**) Safety factor diagram, minimum value 12.4.

**Figure 7 polymers-14-00352-f007:**
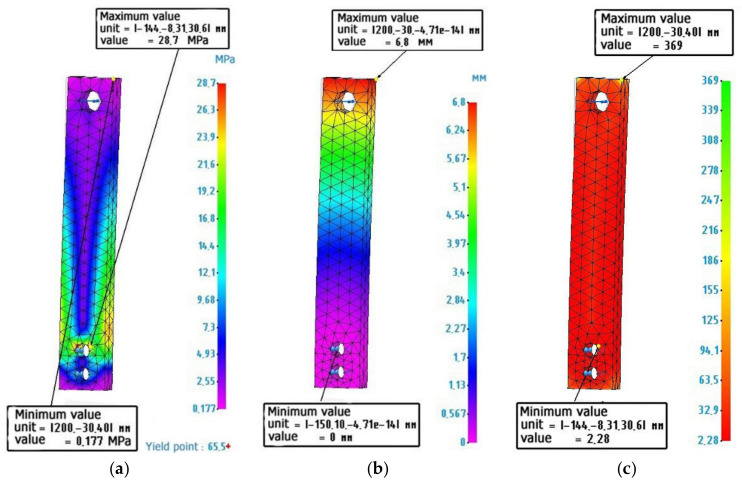
Finite element analysis of a metal-polymer sample: (**a**) Stress diagram according to von Mises, maximum value 28.7 MPa; (**b**) Strain diagram, maximum value 6.8 mm: (**c**) Safety factor diagram, minimum value 2.28.

**Figure 8 polymers-14-00352-f008:**
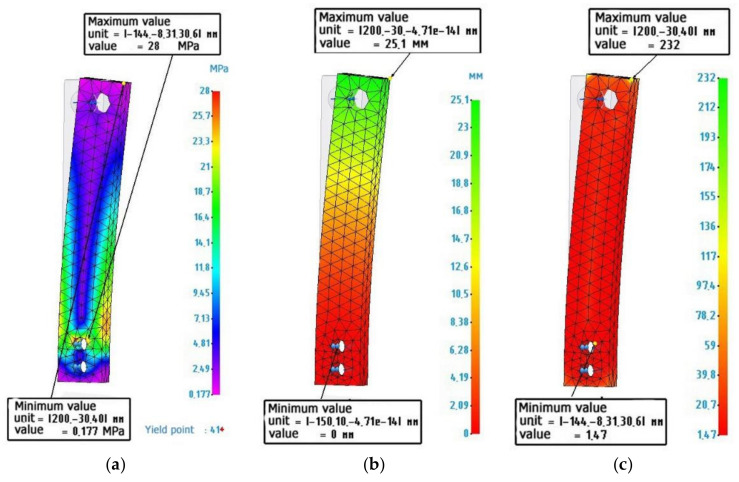
Finite element analysis of an ABS plastic sample: (**a**) Stress diagram according to von Mises, maximum value 28 MPa; (**b**) Strain diagram, maximum value 25.1 mm: (**c**) Safety factor diagram, minimum value 1.47.

**Figure 9 polymers-14-00352-f009:**
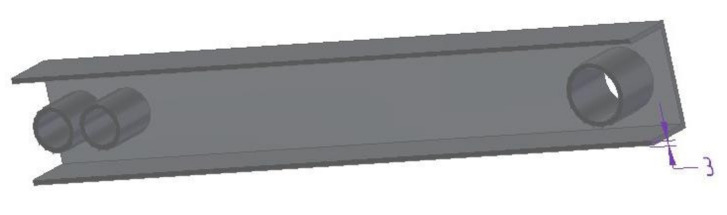
Longitudinal section of the metal shell.

**Figure 10 polymers-14-00352-f010:**
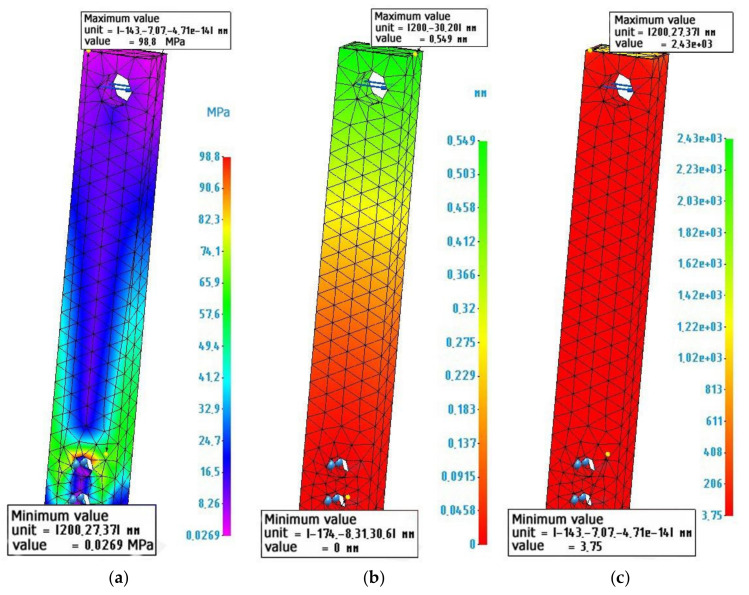
Finite element calculation of a metal-metal-polymer composite sample: (**a**) Stress diagram according to von Mises, the maximum value is 98.8 MPa; (**b**) Strain diagram, maximum value 0.549 mm: (**c**) Safety factor diagram, minimum value 3.75.

**Figure 11 polymers-14-00352-f011:**
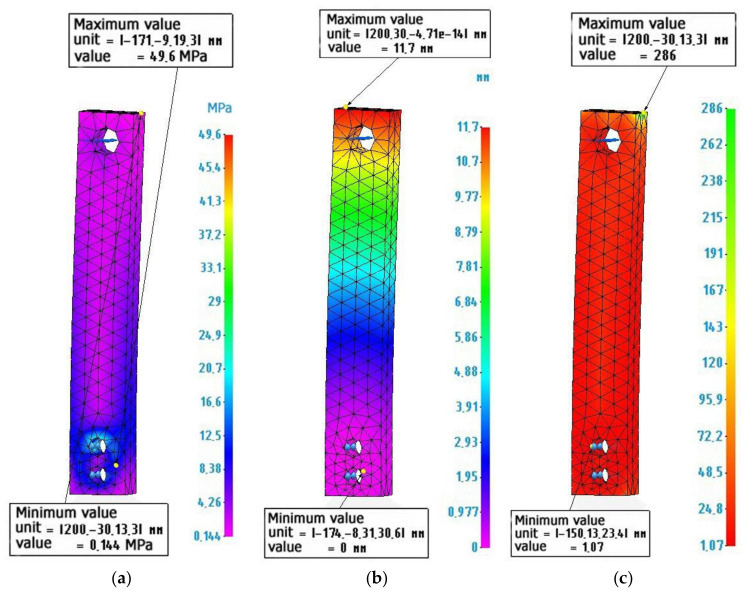
Finite element calculation of a composite sample ABS plastic-metal-polymer: (**a**) Stress diagram according to von Mises, maximum value 49.6 MPa; (**b**) Strain diagram, maximum value 11.7 mm: (**c**) Safety factor diagram, minimum value 1.07.

**Figure 12 polymers-14-00352-f012:**
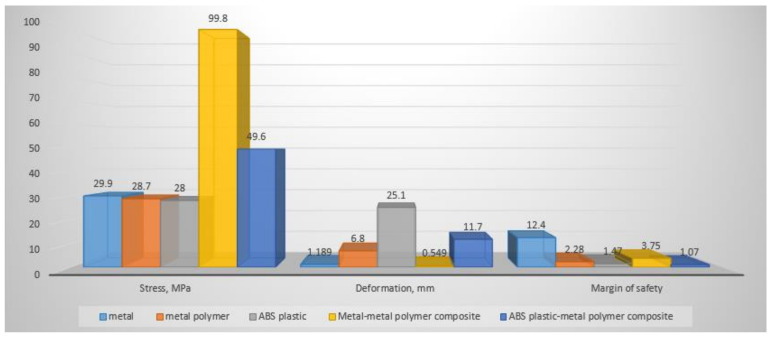
Comparison of strength characteristics of specimens under a load of 2000 N manufactured using various technologies.

**Figure 13 polymers-14-00352-f013:**
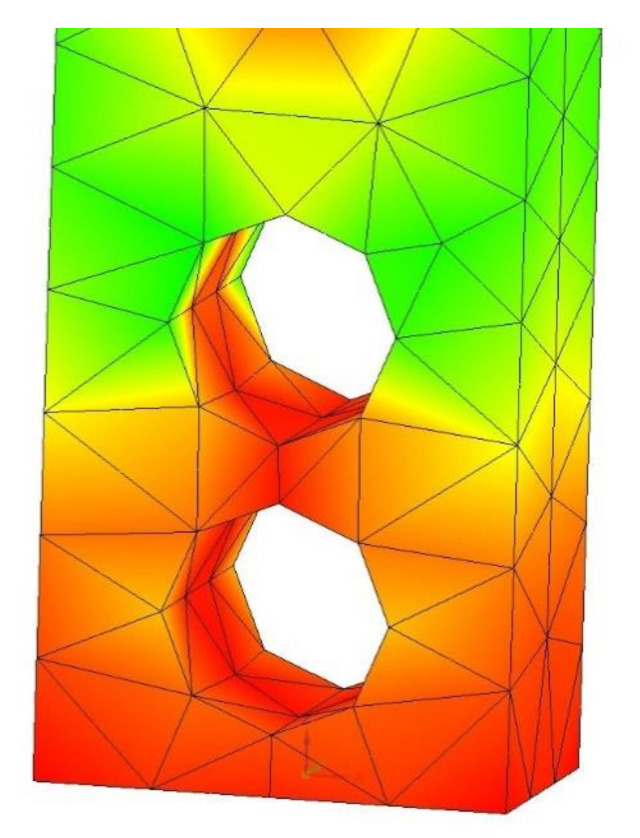
Stress concentrator on the surface of the mounting hole.

**Figure 14 polymers-14-00352-f014:**
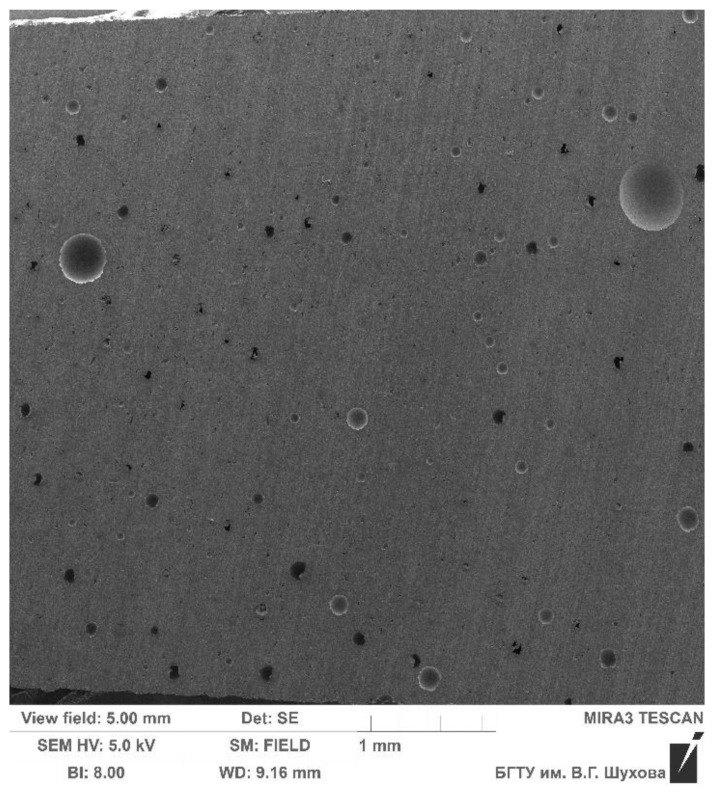
An enlarged cut of metal-polymer material “LEO filled with aluminum” cured at atmospheric pressure.

**Figure 15 polymers-14-00352-f015:**
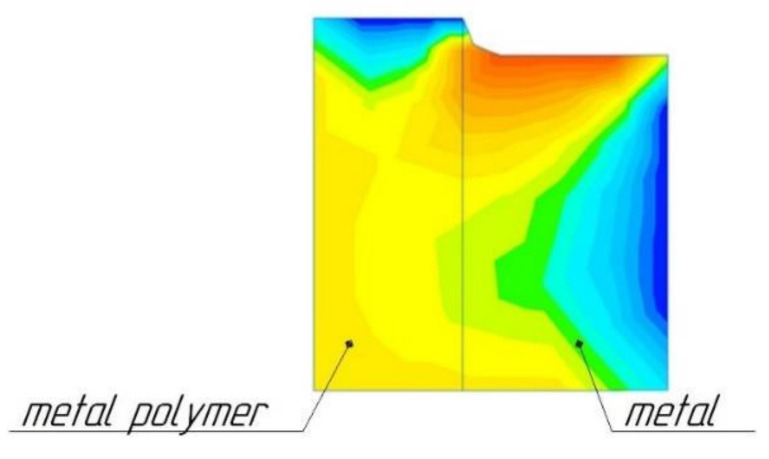
Diagram of the distribution of temperature fields from metal to metal-polymer during grinding, Reprinted from [18], Taylor and Francis group, Copyrights 2022, UK.

**Table 1 polymers-14-00352-t001:** Cost data for 3D printing stainless steel.

Source	Cost 1 cm^3^ from, $
Sprint3d technologies [6]	12.59
Cubic prints 3D Printing Service [7]	11.19
Studia3d 3D Printing Service [8]	14.30
Mg3d 3D Printing Service Homepage [9]	12.59
Center of technological competence of additive technologies [10]	13.29
Mean	12.79

**Table 2 polymers-14-00352-t002:** Properties of metal-polymer materials.

Commercial Name	Compressive Strength (DIN 53281-83), MPa	Tensile Strength (DIN 53281-83), MPa	Flexural Strength (DIN 53281-83), MPa	Mixture Viscosity, mPa × s	Young’s Modulus in MPa	Source
WEICON WR Liquid, Steel-Filled	110	33	80	20,000	5500	[21]
Devcon Plastic Steel Liquid (B)	70	-	-	25,000	-	[22]
Metal Polymer LEO “Ferro-khrom”	230	-	76	-	6000	[23]
Loctite Hysol 3479	90	60	-	-	6000	[24]

**Table 3 polymers-14-00352-t003:** Characteristics of the materials used in the calculations.

Material	Density, Kg/m^3^	Elastic Modulus, MPa	Poisson’s Number	Yield Stress, MPa	Strength Limit, MPa
Stainless steel (metal)	7830	215,000	0.3	370	610
Metal polymer «Ferro-chromium»	2500	6000	0.37	65.5	230
ABS plastic BESTFILAMENT	1050	1627	0.4	41	22

## Data Availability

The data presented in this study is available upon request from the respective author. The data are not publicly available as they are also part of ongoing research.

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
