# Peer review of "Justification of the Use of Composite Metal-Metal-Polymer Parts for Functional Structures"

_polymers, 2022, doi:10.3390/polym14020352_

Round 1
Reviewer 1 Report
- Research objectives are generic. It should be specific.
- Please don’t use Abbreviation without its expanded form.
- There is no validation of finite element modelling in this study.
- Convergence study is mixing.
Author Response
The team of the article “Substantiation of the efficiency of manufacturing a composite metal-metal-polymer functional part instead of a fully printed metal on a 3D printer, based on a comparison of their strength characteristics” thanks the reviewers for their professional and relevant comments.
Comments on reviewer 1 сomments:
- The purpose of the study was rewritten specifically and in more detail.
- The decodings of abbreviations have been added:
ABS (acrylonitrile butadiene styrene)
FDM (fused deposition modeling)
SWOT (Strengths, Weakness, Opportunities, Threats)
EASC (the Euro-Asian Council for Standardization, Metrology and Certification)
- All models and assemblies indicating the used simulation parameters were made publicly available at the link:
https://drive.google.com/drive/folders/1Vv2h1FlVJ8Gjp2g7W3irrQt9L6TVS53r?usp=sharing
All users can make their own simulation and check the correctness of the presented data.
- The convergence estimate serves as a guide when assigning a grid step depending on the desired accuracy. When solving any sufficiently important problem, one cannot do without analyzing the quality of the solution, which can be verified by re-examining the problem on a different grid of elements. In the presented work, we preliminarily checked the convergence of the results for various sizes of the generation parameters of the tetrahedral finite element mesh. The optimal parameters accepted in the calculations are indicated in the SolidEdge simulator
https://drive.google.com/drive/folders/1Vv2h1FlVJ8Gjp2g7W3irrQt9L6TVS53r?usp=sharing.
The authors deliberately did not give the calculations of the convergence of the results, since this problem relates to the methodology for performing calculations by the finite element method and is indirectly related to the objectives of this study.

Reviewer 2 Report
The article "Calculated justification for the use of composite metal-metal-polymer parts in the manufacturing of functional structures" presents the justification for the use of composite metal-metal-polymer parts for functional structures rather than using additive manufacturing techniques. This article is written in very poor English. Some sentences are very long, convoluted, and make no sense. The manuscript should not be p[ublished in the current form.

Author Response
The team of the article “Substantiation of the efficiency of manufacturing a composite metal-metal-polymer functional part instead of a fully printed metal on a 3D printer, based on a comparison of their strength characteristics” thanks the reviewers for their professional and relevant comments.
Comments on reviewer 2 сomments:
- The decoding of the abbreviation SWOT (Strengths, Weakness, Opportunities, Threats) has been added.
- The duplicate word has been removed.
- The description of the exoskeleton model has been added.
- The decoding of the abbreviation has been added.
- All figure captions have been translated into English.
- In fact, the loads are different, but in hundredths and thousandths of MPa. The number represented in Pa would reflect the difference, but would be too big to represent. The reader can independently experiment with the display of values, as well as check the reliability of the data presented, using the calculation models and simulations provided by the authors:
https://drive.google.com/drive/folders/1Vv2h1FlVJ8Gjp2g7W3irrQt9L6TVS53r?usp=sharing
- The discussion on safety factors has been expanded and supplemented to provide clarity.
- All figure captions have been edited.
- The convergence estimate serves as a guide when assigning a grid step depending on the desired accuracy. When solving any sufficiently important problem, one cannot do without analyzing the quality of the solution, which can be verified by re-examining the problem on a different grid of elements. In the presented work, we preliminarily checked the convergence of the results for various sizes of the generation parameters of the tetrahedral finite element mesh. The optimal parameters accepted in the calculations are indicated in the SolidEdge simulator
https://drive.google.com/drive/folders/1Vv2h1FlVJ8Gjp2g7W3irrQt9L6TVS53r?usp=sharing.
The authors deliberately did not give the calculations of the convergence of the results, since this problem relates to the methodology for performing calculations by the finite element method and is indirectly related to the objectives of this study.
- The mistake has been corrected.
- Figure 14 was obtained by the author in person at the laboratory of the Center for High Technologies (CHT) BSTU named after V.G. Shukhov, as mentioned in the Acknowledgments.
Figure 15 is taken from [13], this link has been added to the manuscript.
- It is not entirely clear what data, in the opinion of the reviewer, should be added, but the authors would like to focus on the following own data:
In Section 4.1, the authors analyzed their own data of finite element calculations, on the basis of which conclusions were drawn about the relevance of using a composite structure of a part made of a metal shell filled with a metal polymer;
In section 4.2, based on the personal experience of the authors in the production of metal-metal-polymer resins, the main problems that exist today in the development of technology for obtaining composite parts of the proposed design are described.
13. The purpose of the study has been described in more detail. Perhaps earlier it was not entirely clear what results were planned to be obtained. The report has been revised to reflect a comparison between the proposed method for producing a composite part consisting of a metal shell and a metal-polymer filler and an all-metal part.

Reviewer 3 Report
Some general comments:
The paper needs extensive editing of the English language.
The title of the article must be changed as it does not seem to convey its purpose. I don’t like the words“ Calculated justification…”
Some line by line comments:
Line 23 “The article provides a rationale for the main problems facing researchers and technologists in the development…”
This sentence is not clear
Line 32 “...definition of GOST 14.201-83…”
This is a Russian standard. Maybe the authors could use an International standard.
Line 75 “...inexpensive 3D FDM and SLA prints. As…
The reader does not know what this acronym is (SLA)
Line 82 …since in serial and mass production, this technology will obviously lose out to the classical technology of molding thermoplastic polymers into metal molds.”
This sentence is not clear
Line 123 “… calculated justification of the strength characteristics of composite parts based on a metal-polymer…”
I don’t like the words “...calculated justification…”
Line 151 “...In the study of a composite part made of a thin-walled shell with a thickness of 2 mm and a filler in the form of a metal-polymer composition, in order to reduce the computing power of a computer, a model of the sample presented in Figure 2 has been developed.”
This sentence is not clear.
Line 157 “...If the manufacture of composite parts for an exoskeleton is taken, then carrying out strength calculations should be carried out for loads that can be tested by a device (exoskeleton) when performing its service purpose.”
This sentence is not clear.
Line 173 The figures 3 to 11 have the subtitles in russian. Must be in english.
Line 180 “...action of the created torque…”
Torque or bending?
Line 198 “…be loaded by half more than the current load and this will not lead to its…”
How is it possible to have a safety margin of 0.426 and say “loaded by half more…”? How did the authors calculate the safety margin?
Line 246 The figure is number 9, not 10.
Line 259 The figure 12 has some errors. The stress of the metal-metal polymer composite is 98.8 MPa and the deformation of metal is 0,189 mm.
Line 265 “...When designing a metal shell, it is necessary to provide for the reinforcement of the structure of the metal shell at the points of contact due to the construction of additional supports from the shell material….”
The sentence is not clear.
Line 277 “...filler is the least susceptible to deformation, which…”
The deformation of metal (0,189 mm) is less than metal-metal polymer (0,549 mm)
Line 285 “...part; the possibility of a civilian exoskeleton in the design is unlikely…”
The sentence is not clear
Line 312 Figure 14, not figure 1.
Line 345 “...heated metal in contact with a metal pol-345 ymer can lead to thermal destruction of the latter, since…”
The sentence is not clear
Author Response
The team of the article “Substantiation of the efficiency of manufacturing a composite metal-metal-polymer functional part instead of a fully printed metal on a 3D printer, based on a comparison of their strength characteristics” thanks the reviewers for their professional and relevant comments.
Comments on reviewer 3 сomments:
1) The English language has been further revised.
2) The title of the article has been changed to “Substantiation of the efficiency of manufacturing a composite metal-metal-polymer functional part instead of a fully printed metal on a 3D printer, based on a comparison of their strength characteristics”
3) The proposal has been reformulated for better understanding.
4) A reference to EASC is given to describe the definition of the manufacturability of the structure. The use of a Russian o international standard is of no fundamental importance in this case.
5) The decoding of the abbreviation has been added. SLA (Stereolithography Apparatus)
6) The wording has been changed.
7) The expression "... calculated justification ..." has been reformulated.
8) The paragraph has been rewritten in a different interpretation.
9) The paragraph has been rewritten in a different interpretation.
10) Figure captions have been corrected to English.
11) The authors meant not 2 times, but 0.5 times. Unfortunately, the idea was not conveyed here due to incorrect translation.
12) The mistake has been corrected.
13) The authors rechecked the data presented in figure 12 with the calculation data for the metal-metal polymer sample (figure 10) and the calculation data for the metal sample (figure 6). The data is presented correctly.
14) The wording has been corrected for better understanding.
15) Indeed, there is a mistake in figure 12. Corresponding corrections in the description and figure 12 have been made.
16) The paragraph has been rewritten in accordance with the recommendations of all reviewers.
17) The corresponding corrections have been made.
18) The paragraph has been rewritten in an attempt to make the meaning clearer.

Round 2
Reviewer 1 Report
Authors have carried out revision of Comments 1 & 2. But justification for not performing convergence study and validation of results are not acceptable.
Reviewer 2 Report
Please see attached.

Round 3
Reviewer 1 Report
Authors have carried out revision satisfactorily.